# Bioengineering Human Upper Respiratory Mucosa: A Systematic Review of the State of the Art of Cell Culture Techniques

**DOI:** 10.3390/bioengineering11080826

**Published:** 2024-08-13

**Authors:** Davaine Joel Ndongo Sonfack, Clémence Tanguay Boivin, Lydia Touzel Deschênes, Thibault Maurand, Célina Maguemoun, François Berthod, François Gros-Louis, Pierre-Olivier Champagne

**Affiliations:** 1Department of Surgery, Faculty of Medicine, Laval University, Quebec, QC G1V 0A6, Canada; clemence.tanguay-boivin.1@ulaval.ca (C.T.B.); thibault.maurand.1@ulaval.ca (T.M.); celina.maguemoun.1@ulaval.ca (C.M.); francois.berthod@fmed.ulaval.ca (F.B.); francois.gros-louis@fmed.ulaval.ca (F.G.-L.); pierre-olivier.champagne.2@ulaval.ca (P.-O.C.); 2LOEX, CHU de Québec—Laval University Research Center, Quebec, QC G1J 5B3, Canada; lydia.t-deschenes@crchudequebec.ulaval.ca; 3Laval University Neurosurgery Innovation Laboratory (LINUL), Quebec, QC G1J 5B3, Canada; 4Department of Neurosurgery, Faculty of Medicine, Laval University, Quebec, QC G1V 0A6, Canada

**Keywords:** bioengineering, endoscopic endonasal surgery, nasal mucosa, regenerative medicine, tissue engineering

## Abstract

Background: The upper respiratory mucosa plays a crucial role in both the physical integrity and immunological function of the respiratory tract. However, in certain situations such as infections, trauma, or surgery, it might sustain damage. Tissue engineering, a field of regenerative medicine, has found applications in various medical fields including but not limited to plastic surgery, ophthalmology, and urology. However, its application to the respiratory system remains somewhat difficult due to the complex morphology and histology of the upper respiratory tract. To date, a culture protocol for producing a handleable, well-differentiated nasal mucosa has yet to be developed. The objective of this review is to describe the current state of research pertaining to cell culture techniques used for producing autologous healthy human upper respiratory cells and mucosal tissues, as well as describe its clinical applications. Methods: A search of the relevant literature was carried out with no time restriction across Embase, Cochrane, PubMed, and Medline Ovid databases. Keywords related to “respiratory mucosa” and “culture techniques of the human airway” were the focus of the search strategy for this review. The risk of bias in retained studies was assessed using the Joanna Briggs Institute’s (JBI) critical appraisal tools for qualitative research. A narrative synthesis of our results was then conducted. Results: A total of 33 studies were included in this review, and thirteen of these focused solely on developing a cell culture protocol without further use. The rest of the studies used their own developed protocol for various applications such as cystic fibrosis, pharmacological, and viral research. One study was able to develop a promising model for nasal mucosa that could be employed as a replacement in nasotracheal reconstructive surgery. Conclusions: This systematic review extensively explored the current state of research regarding cell culture techniques for producing tissue-engineered nasal mucosa. Bioengineering the nasal mucosa holds great potential for clinical use. However, further research on mechanical properties is essential, as the comparison of engineered tissues is currently focused on morphology rather than comprehensive mechanical assessments.

## 1. Introduction

The upper respiratory tract extends from the nose to the larynx and functions primarily to filter and transport oxygen-rich air to the lungs [1]. The nasal mucosa is comprised of goblet, ciliated, and basal cells which function all together to ensure physical and immunological barrier against dust and pathogens [1,2] (Figure 1). Goblet cells secrete mucus which traps dust and foreign particles from the airways preventing them from reaching the lungs [3]. Ciliated cells have hair-like projections that oscillate to push mucus and trapped particles toward the throat where they can be expelled from the body [3]. Attached to the basement membrane by hemidesmosomes, basal cells serve as a physical barrier that separates the nasal cavity from the internal body [3]. These cells also display stemness properties as they can differentiate into goblet or ciliated cells to replace damaged ones [3]. The spatial arrangement of these cells provides the nasal mucosa with its distinctive pseudostratified ciliated columnar epithelium [4]. All the cells in this epithelium are in direct contact with the basement membrane, resulting in a single layer of cells with varying thicknesses [4]. Therefore, the nasal mucosa plays a crucial role in proper breathing by channeling, filtering, and humidifying the inspired air before it reaches the lungs [5,6]. These functions are essential as they act as key initial steps of our immune system, preventing harmful pathogens or very cold air from reaching and disrupting alveoli homeostasis in the lungs [7,8].

Stresses to the nasal epithelium such as infections, traumas, and surgeries, create a potential risk for the patient and, therefore, require restoration of its structural integrity [2,9]. For instance, allergens containing proteases in conditions like allergic rhinitis, along with diesel exhaust particles and cigarette smoke, pose potential risks that can compromise the integrity of the respiratory epithelial barrier [10,11,12,13,14]. When the nasal mucosa’s sealing is compromised due to inflammation, these particles infiltrate the submucosal space, leading to infections and hindering its effective functioning [2]. Also, surgical procedures performed within the nasal cavity, such as endoscopic endonasal skull base surgeries, intranasal tumor resection surgeries, or functional endoscopic sinus surgeries, often necessitate the intentional disruption of the nasal epithelium as part of the medical intervention [15,16,17]. Consequently, there is a significant interest in thoroughly investigating the healing process and exploring current techniques available for repairing the damaged respiratory epithelium.

Tissue engineering is a field of regenerative medicine that aims to produce bioengineered constructs for medical purposes. The goal is to develop functional biomaterials capable of restoring, maintaining, or repairing damaged tissues in the human body [18]. For example, in patients with skin burns where significant tissue has been lost, tissue engineering focuses on developing bioequivalent skin grafts that can be used as transplants to replace the lost tissue [19,20]. In other fields, such as genetic disorders, tissue engineering is valuable for creating personalized disease models including those used to study cystic fibrosis, dystrophic epidermolysis bullosa, neurofibromatosis type 1, aortic valve calcification, cancers and amyotrophic lateral sclerosis [21,22,23,24,25,26,27,28,29,30]. These models facilitate the study of immune reactions to diseases, characterize the morphological and histological changes in nasal tissue caused by these conditions, and explore potential drug therapies [21]. Finally, in addition to providing bioequivalent tissue for transplants and disease models, tissue engineering is also utilized to study drug interactions with nasal tissue and to investigate nasal mucosa adaptations to viral infections [31,32,33,34]. Therefore, employing tissue engineering to reconstruct the nasal mucosa with autologous (harvested from the patient) upper respiratory cells emerges as a promising avenue for studying the respiratory healing process, and eventually creating a handleable in vitro nasal mucosa for clinical applications. Advancements in tissue engineering and 3D cell culture techniques open the possibility of generating autologous tissues through regenerative medicine. 

The current state of research on tissue engineering and its applications on the upper respiratory tract is not well defined. The complexity of the respiratory system has led to a diversity of protocols and methodologies in the literature for generating nasal mucosa. This heterogeneity makes it challenging to identify which approaches are easily reproducible and suitable for producing functional respiratory mucosa. This review aims to bridge this knowledge gap by conducting a comprehensive and systematic review of the literature pertaining to cell culture techniques used for producing autologous healthy human upper respiratory mucosal tissue and its applications. 

## 2. Materials and Methods

The preferred reporting items for systematic reviews and meta-analysis statement (PRISMA) guidelines and checklist were followed during the elaboration of this review [35]. The PRISMA checklist is available as Appendix A. The protocol for this systematic review has been registered in PROSPERO [36] (ID CRD42022368121). This systematic review was only based on the published literature and did not necessitate patient recruitment. 

### 2.1. Eligibility Criteria 

To be eligible for this systematic review, studies had to describe a comprehensive cell culture protocol for producing an in vitro engineered tissue using human upper respiratory cells. Studies were included in the Covidence systematic review software (Veritas Health Innovation, Melbourne, Australia) [37]. No time restriction was applied during the search and all study designs as well as data from grey literature (companies’ and preprint publications) were included. During the screening process, only studies written in English and French were retained. Studies without a cell culture protocol or using non-human cell culture protocols were excluded. Studies describing protocol using spheroids, lung-on-a-chip models, or studies with the primary goal of generating cells from the lower respiratory system were excluded. The exclusion of these studies stems from the primary aim of this review, which is to describe cell culture protocols employing in vivo nasal tissue biopsies. In doing this, it amplifies the specificity and relevance of the review’s findings within a clinical context.

### 2.2. Search Strategy 

A literature search was carried out across Medline Ovid, Embase, PubMed and Cochrane on 7 November 2023. The search was performed using keywords and Mesh terms related to culture techniques [MESH] and respiratory mucosa [MESH]. These concepts were linked using the boolean operator AND. We also searched titles, abstracts, and keywords sections for synonyms of the MESH terms. Cell culture, cell tissue culture, respiratory mucosa, nasal mucosa, and respiratory epithelium were the synonyms searched. The search strategy as well as the results from Medline, Embase and PubMed can be found in Appendix B, Appendix C, and Appendix D, respectively. The Cochrane search yielded no result. The search strategy was verified by an external medical reference librarian. 

### 2.3. Data Management and Risk of Bias Assessment 

The manuscripts yielded by the search results were exported to Covidence for screening by two independent reviewers following the inclusion and exclusion criteria. Full text screening of selected articles was subsequently performed to reevaluate their eligibility for review. Any disagreements between these two were resolved by a third independent screener. A codification guide was made available for extraction, which was also performed by two independent reviewers. 

An overview of the Tissue Engineering protocol is represented in tabular form in Figure 2. The following data was extracted from each selected full-text study: article title, first author, year of publication, journal of publication, population, total sample size, types of samples collected, method for cell collection, cell line, cell culture medium, antibiotics protocol, cell digestion enzyme protocol, extraction protocol, cell culture method, cell identification protocol, quality assessment of sample and results. The primary objective was to determine if the authors were able to produce an upper human respiratory mucosa that was histologically and functionally comparable to the native tissue. Data on the quality and viability of the generated tissue were gathered. Data on study limitations and causes of failure were also collected. 

Risk of bias in individual studies was assessed using the Joanna Briggs Institute (JBI) Critical Appraisal tools for Qualitative Research [38]. This checklist contains 10 questions that assess specific domains of studies to determine the potential risk of bias and could be answered with ‘yes’, ‘no’ or ‘unclear’. The JBI checklist can be found in Appendix E. Any disagreements between reviewers were discussed and resolved by consensus. The risk of bias of individual studies was assessed using the following cutoffs: low risk if 30% or less of the questions scored “no”, moderate risk if 30 to 60% of questions scored “no”, and high risk if “no” scores were above 60% [39,40]. 

Furthermore, two independent reviewers were involved in the screening process as a means of limiting the risk of selection bias. Moreover, we screened the literature with no time interval nor journal impact factor restriction as a way of diminishing the impact of publication bias in this review. A narrative synthesis of the results was performed as the main outcome was qualitative. 

## 3. Results

### 3.1. Study Selection

The search strategy yielded a total of 4193 articles from all databases. From those, 647 duplicates were removed (Figure 3). The search in grey literature yielded no articles. During the title and abstract screening process, 3236 records were excluded and 310 were retained for full-text assessment. Of those, 277 were disqualified as they did not meet the inclusion criteria. Therefore, a total of 33 studies were included in this review. 

### 3.2. Tissue Engineering Protocol Overview 

Published protocols for the differentiation of human nasal epithelial cells (HNEC) at the air-liquid interface (ALI) commonly follow similar procedures. These generally include acquiring HNEC through diverse methods, extracting viable cells from HNEC using a variety of enzymes, cultivating and expanding the collected cells, and then transferring them to porous cell culture inserts for differentiation at the ALI. Nevertheless, differences can be found among protocols regarding the method of cell collection, the types of differentiation and proliferation media employed, the inclusion of various media supplements and antibiotics, the frequency of cell passaging, and the characterization of the resulting tissue-engineered constructs. Cell passaging, also known as subculturing or splitting, refers to the process of transferring cells from one culture vessel to another in order to maintain healthy and actively growing cell populations [41]. 

### 3.3. Cell Collection and Intended Use

Cell source, collection method and population type: A summary of collected data regarding cell source and collection method from all included articles is detailed in Table 1. With the objective of generating tissue-engineered nasal mucosa, 19 studies used nasal cells [21,31,32,42,43,44,45,46,47,48,49,50,51,52,53,54,55,56,57], 3 used bronchial cells [58,59,60], 10 employed a combination of both [22,33,34,61,62,63,64,65,66,67] and 1 utilized primary basal epithelial cells extracted from the nasal epithelium [68]. Not all the studies included in the analysis provided information about the patient population from which cells were collected [21,22,31,32,42,44,45,48,50,52,53,54,55,57,58,59,62,66,68]. However, among those that did specify it, cells were obtained from cadavers in six studies [34,60,61,64,65,67], from adult patients in five studies [43,46,49,51,63], and from pediatric patients in three studies [33,47,56]. Moreover, the method of HNEC collection varied significantly across studies. Nasal biopsy emerged as the preferred method, employed by nine studies [32,42,44,50,51,52,53,56,57], followed by nasal brushing (*n* = 8) [21,22,43,45,46,47,49,55], bronchial biopsy (*n* = 8) [34,48,60,61,62,64,66,67], bronchial brushing (*n* = 3) [33,44,59], and, lastly, the utilization of lung explants (*n* = 2) [21,22] for collecting nasal cells. Regarding cell collection methods, It is important to note that the quantity of cells obtained through nasal brushing is consistently lower than that obtained through a conventional biopsy, and would, therefore, necessitates a thorough and prolonged brushing of the mucosa to obtain a higher number of HNEC [43]. None of the studies included in the analysis quantitatively compared the cell yield between brushing and biopsy procedures. Stokes et al. examined nasal cell isolation employing three distinct brushing techniques: cytology brush, swab, and curette [69]. They observed that the highest success rate in establishing primary cell cultures was achieved with brushes: 90% compared to 65% for swabs and 70% for curettes [69]. While there has not been any research conducted on brushing vs biopsy, the general consensus is that invasive procedures like tissue biopsy are presumed to yield more cells available for extraction compared to non-invasive methods like nasal brushing, likely due to the larger sample size. Nevertheless, the brushing method may be more suitable for specific populations, such as pediatric cases, due to its less invasive nature and can still provide successful differentiated ALI cultures [43,47]. Concerning the population from which cells are extracted, if HNEC are to be obtained from cadavers, the post-mortem interval prior cell extraction should be as low as possible and must be performed within 8 h of death to increase the likelihood of obtaining the highest number of viable cells in the biopsy [54,64]. 

Intended use: Among the included studies, 13 described a protocol for upper respiratory cell culture without further use [43,45,50,52,56,58,59,60,62,63,64,65,67]. Other studies developed a respiratory cell protocol for diverse purposes. Some investigated cystic fibrosis disease (n = 5) [21,22,47,54,66], explored the function and biological processes within the nasal mucosa (n = 5) [44,46,51,61,68], studied drug transport and metabolism (n = 4) [31,32,49,57], explored nasal tissue engineering applications for reconstructive surgery (n = 4) [42,48,53,55] and studied viral infections of the nasal mucosa (n = 2) [33,34]. In vitro tissue models for viral research are usually developed to study how the nasal mucosa adapts to viral infections [33,34]. 

### 3.4. Cell Culture Protocol

Antibiotics and enzyme protocol: Regarding cell culture protocol, a combination of penicillin and streptomycin were the most used antibiotics to prevent infections from the donor tissue among the included studies (n = 24), Table 2. Six studies added gentamycin to their penicillin/streptomycin mixture [44,48,54,64,67,68] and one used gentamycin alone [60]. Seven studies did not provide an antibiotic protocol [31,42,46,50,52,53,61]. Considering the nasal cavity’s high susceptibility to bacterial colonization, the risk of infection and contamination in cell culture is markedly elevated. A common observation identified by most of the included studies was sample contamination, highlighting the importance of using a robust antibiotic combination to prevent bacterial contamination. To ensure an even greater coverage against pathogens, some authors also supplemented their medium with amphotericin B to prevent infections from yeasts and fungi [21,22,48,64]. In terms of human nasal tissue digestion and cell extraction, following biopsies, the most used enzyme was protease (n = 10) [21,51,53,54,57,61,62,64,66,67] followed by pronase (n = 6) [22,32,34,52,55,61], DNase (n = 3) [50,61,64] and collagenase (n = 3) [42,53,59]. Fifteen studies did not mention the enzyme being utilized for cell extraction. 

Submerged culture versus air-liquid interface cultures: Submerged culture involves the complete cultivation of cells in a liquid medium, while air-liquid interface (ALI) cultures promote more terminal differentiation of epithelial cells. This approach is designed to reproduce the physiology of the nasal mucosa, where the outer membrane is in contact with air, and the inner membrane interacts with the internal body. In the reviewed studies, the predominant method for cell culture was the air-liquid Interface (n = 26), reflecting an emphasis on obtaining a three-dimensional model that resembles and functions like the native respiratory tissue. However, some studies chose a 2D submerged cell culture approach either because it better suited their research objectives or because it was the most advanced culture option available at the time (n = 7) [32,49,53,56,57,60,67]. Therefore, choosing a 3D culture method aims to obtain in vitro tissue-engineered constructs that are functionally and morphologically similar to the native tissue. This will facilitate the transfer of the downstream in vitro research results and their clinical application. 

Culture media and supplements: Table 2 provides a comprehensive description of the medium used by each study to ensure the proliferation and expansion of the cells. Dulbecco’s Modified Eagle Medium (DMEM) emerged as the most frequently employed medium in the included studies for human nasal epithelial cell growth [33,42,45,48,55,56,57,67,68]. Additionally, some studies opted for DMEM/F12 [22,34,44,51,53,59], an enriched version of DMEM supplemented with Ham’s nutrient mixture F12, while others supplemented their DMEM with Defined Keratinocytes Serum-Free Medium (DKSFM) to enhance keratinocyte growth [42,53]. Notably, one study favored Minimum Essential Medium (MEM) [66], a less nutrient-rich alternative to DMEM for their cell expansion. Different media can be incorporated with DMEM as evidenced by one study in which DMEM was combined with LHC-9 medium [67]. Notably, certain studies preferred alternative media like Bronchial Epithelial Growth Medium (BEGM) [32,46,47,48,50,52,54,58,61,62,64,65,68], Airway Epithelial Cell Growth Medium (AECGM) [31,49] and Pneumacult [43]. In studies where cells were grown using the air-liquid interface, DMEM (n = 10) [21,22,42,44,52,54,59,64,65,66] and Pneumacult (n = 7) [34,43,45,47,50,61,68] stood out as the most frequently utilized medium for further differentiating cultured cells into a mucociliary respiratory epithelium. This was followed by BEGM (n = 5) [45,48,58,61,62], AECGM (n = 4) [31,33,44,45], LHC-basal medium (n = 3) [52,54,64] and Bronchial-ALI (n = 3) [46,51,63]. Regarding media supplementation with serum, Fetal Bovine Serum (FBS) was the most utilized serum in the included studies (n = 13) [22,33,34,42,44,45,48,51,53,56,57,59,62]. Other studies (n = 7) favored Bovine Serum Albumin (BSA) which is a single protein purified from bovine serum [32,43,54,58,64,67,68]. One study opted for Adult Bovine Serum (ABS) which is derived from cows aged twelve months or more [21]. Notably, six studies [49,50,52,55,60,61] did not use serum to supplement the culture media, and an additional six did not provide information about the serum content in their media [31,46,47,63,65,66]. Given that some groups seek to obtain entirely human cultures for clinical applicability reasons, they choose not to supplement their culture medium with animal serum and develop their own serum-free medium [22,70].

Regarding cell culture protocol among the included studies, Pneumacult was the media that appeared the most suitable for the mucociliary differentiation of cultured cells. Indeed, studies noted that cells grown in Pneumacult exhibited more cilia and more mucus-secreting cells compared to BEGM and AECGM [45,61]. Lee et al. found that MUC5AC was significantly more upregulated in Pneumacult than in the other media (*p*-value < 0.0001) [45]. These findings were supported by another study that found that Pneumacult cultures had significantly increased proportions of ciliated cells (70.0 ± 2.0%) compared to BEGM (51.4 ± 3.3%, *p* < 0.0001) [61]. Moreover, in the same study, Pneumacult cultures were found to express 2.3 times the protein MUC5AC compared to BEGM (*p* < 0.05) [61]. Moreover, there is no consensus among studies regarding media supplementation with serum. While some successfully achieved well-differentiated nasal mucosa using serum-enriched media, others strongly discourage its use, citing experiments where better-differentiated cells were grown in serum-free media [21,59,60,61]. On one hand, Gianotti et al. argued that serum-free media produces a well-differentiated epithelial monolayer, which proves valuable for assessing the morphological properties of primary airway epithelia [21]. This includes a pseudo-stratified epithelium with an increased number of ciliated and goblet cells observed by immunofluorescence [21]. No quantitative data was provided to support their findings. On the other hand, serum-enriched media are more suitable for functional electrophysiologic assays. However, a serum-free approach comes with the trade-off of requiring an extended culture time and additional costs related to media supplement replacement [21]. On a related note, retinoic acid is a medium supplement that multiple studies have identified as promoting ciliogenesis [42,59,60,61]. Therefore, these studies highly recommend incorporating retinoic acid into the medium to ensure proper ciliary differentiation. Furthermore, certain studies conducted co-cultures of HNEC with 3T3-J2 mice fibroblast cells combined with a rho-associated protein kinase (ROCK) inhibitor (Y-27632) [48,68]. As described previously, mice fibroblast feeder cells function as a nourishing layer whereas ROCK inhibitors prevent cell death and facilitate proliferation and expansion [71,72,73]. In one of these studies, the authors found that cells cultured on feeder layers led to the formation of sufficient cell colonies whose morphology did not change with passage [48]. They argued that this coculture method reduces the time required for nasal cell culture and improves cell quality, whereas Zhao et al. found that cell culture quality was not compromised with or without 3T3-J2 cells [51]. The main rationale in favor of avoiding the use of such animal-derived feeder cells is the potential risk of transmitting animal-derived viruses and prions into the culture environment [51].

### 3.5. Cell Characterisation Protocol 

Immunofluorescence protocol and cell identification: As described in Table 3, following cell extraction, 14 studies used antibodies targeted towards MUC5Ac to identify goblet cells by immunofluorescence [21,33,34,43,44,45,46,47,48,51,52,53,61,63]. 3 studies opted for Periodic Acid Schiff (PAS) stain for goblet cell identification [56,59,65]. PAS staining is not an immunofluorescence protocol but rather a staining technique used in histology and cytology to detect carbohydrates, particularly glycogen, glycoproteins, and mucopolysaccharides in tissues [74,75]. Therefore, it is particularly useful in identifying structures like glycogen granules in the liver or mucins in mucus-secreting cells [74,75,76]. Sixteen articles did not perform goblet cell identification. For the identification of ciliated cells, B-tubulin was employed in 7 studies [21,33,34,45,47,51,52], acetylated tubulin in 5 studies [43,44,46,48,63], and cytokeratin-18 in 1 study [53]. Ciliated cell identification was not conducted in 20 studies. For tight junction identification, zonula adherence-1 (ZO-1) was employed in 12 studies [21,33,34,44,46,47,51,52,61,63,65]. Tight junction identification was not conducted in 21 studies. 

Cell viability assessment: As cells undergo extraction, cultivation, and passaging, it becomes crucial to monitor the number of viable cells and assess cellular death. Various methods were employed in some studies to address this aspect. Trypan blue, utilized in six studies [31,33,44,49,57,68], was the most employed procedure for cell counting and viability assessment. Trypan blue, a negatively charged dye, selectively stains cells with compromised cell membranes, thereby indicating cell death [77]. Propidium Iodide, another dye impermeant to living cell membranes [78], was used by two studies for assessing cell viability [34,45]. Additionally, Lactate Dehydrogenase (LDH), a cytoplasmic enzyme serving as a marker for cell death, was employed in 1 study [34]. LDH is released by cells when cellular damage occurs [79]. 

Regarding cell differentiation and viability, cell cultures typically take 7 days to reach confluency after the initial seeding in a transwell insert [33,62]. By the 4th week, using the air-liquid interface method, the cell cultures achieve full differentiation into a pseudostratified epithelium, characterized by the presence of cilia and mucus secretion [33,55,56,57,58,65]. Subsequent loss of differentiation after multiple passages is another limitation identified by some studies. Indeed, it has been observed that cell differentiation regressed after 4 to 5 passages or after 4 to 6 weeks of air-liquid interface culture [32,51,55,56,57,65,67]. This contrasts with the findings of Shagen et al., who reported that some of the cells required 7 weeks to achieve complete differentiation, suggesting that in certain cases, 4 weeks might not be sufficient [46]. Interestingly, Kreft et al. were able to pass their nasal cells up to 12 times before observing a decline in cell differentiation [31]. To evaluate the success of their in vitro cultured model, authors usually determine how accurately it reflects the in vivo tissue. This can be done by immunofluorescence which is one of the most common procedures performed by studies as it enables them to determine whether or not their in vitro model expresses identical proteins to those of the native tissue [21,43,47,65]. Moreover, Transepithelial electrical resistance (TEER) is another test performed by some studies to compare tight junction formation and cellular barrier integrity [21,45,47,61,65]. The identification of ZO-1 offers an additional way to recognize cells with tight junctions [65]. This is especially important in pharmacological studies or studies aimed at evaluating drug transport in which the TEER value of the cultured model must closely relate to that of the native tissue [31,32].

Following cell characterization, the majority of the studies performed experiments to address the underlying research question. No data pertaining to the mechanical testing of the engineered mucosa in comparison to the native tissue was presented. Only morphological comparisons were performed by most studies as described previously. Out of the 33 articles, the production of a complete tissue-engineered mucosa for reconstructive surgery was described in only one study [48]. They described a culture protocol designed to yield ample quantities of clinically viable HNEC for use in tracheal reconstructive surgery [48]. It is well described in the literature that multiple cell passaging of nasal biopsies hinders their capacity for differentiation into a ciliated epithelium and the quantity of cells produced is insufficient for a viable transplant [48,80,81]. In their study, the authors propose a way around this by co-culturing their HNEC with mitotically inactive mouse embryonic fibroblast feeder cells (3T3-J2) along with a rho-associated protein kinase (ROCK) inhibitor (Y-27632). Mice fibroblast feeder cells would serve as a nourishing layer aimed at enhancing cell proliferative potential through the secretion of diverse factors such as Insulin-like Growth Factors (IGFs) [71,72]. Whereas ROCK inhibitors are known to prevent apoptosis and facilitate cell proliferation [73]. They found that cells grown in this manner formed colonies that retained cell-to-cell contact and whose morphology did not change with passage [48]. To test the in vivo applications of their developed protocol and tracheal graft, the authors used a xenogeneic rat engraftment model to repopulate denuded tracheal grafts. They seeded their cultured HNEC on rat tracheal scaffolds and after 5 weeks, they observed complete restoration of an epithelial barrier around the entire circumference of the tracheal grafts [48]. However, no mechanical or resistant testing was performed on the reconstructed graft to compare its physical properties with those of the in vivo tissue. 

### 3.6. Risk of Bias Assessment

All of the 33 studies were evaluated as having a low bias risk [21,31,32,34,42,43,44,45,46,47,48,49,50,51,52,53,55,56,57,58,59,60,61,63,65,67,82], although a small number of bias criteria did not apply to some fundamental and theoretical protocols and were marked as not applicable [22,33,54,62,64,66,68]. The risk of bias in each study is described in Appendix F. This risk level was assessed using the JBI critical appraisal tool for qualitative research [38]. The risk of bias was measured as low when the study reached up to 30% of “no” scores, moderate if the study attained between 30 to 60% of “no” scores, and high if the study obtained more than 60% “no” scores. Therefore, since all the included studies had a low risk of bias, the narrative synthesis and conclusions of this review remained unaffected. 

## 4. Discussion and Conclusions

We reviewed the literature on tissue engineering methods for producing nasal mucosa, with a focus on their clinical applicability. Of the 4193 articles initially included in this systematic review, only 1% passed the selection process. The absence of a cell culture protocol and the use of different cell types other than those from the upper respiratory tract were the main reasons for exclusion among the excluded studies. As this is an emerging field, a standardized protocol for developing nasal mucosa has yet to be established. As illustrated by this review, the applications for such tissue are numerous [21,31,42,44], but lots of challenges are yet to be overcome to produce tissue-engineered autologous nasal mucosa for clinical use. 

Transitioning from a 2D culture to a 3D structure that replicates the architecture and function of the native tissue represents a significant leap forward in the field of tissue engineering. In a 2D culture model, cells are grown as a monolayer on a flat surface such as a petri dish or a culture flask [83,84]. The key benefits of a 2D model include its cost-effective production, easy access to cells, and easy maintenance of the culture [83]. Inexpensive and readily available media such as DMEM can be employed in such situations because the primary objective of the culture is not to replicate an in vivo structure and function. Therefore, the 2D cell culture approach is widely used to determine drug concentrations, evaluate drug toxicity, and preclinical drug candidate testing [84,85,86,87]. One of the main limitations of this cell culture approach is the dedifferentiation and change of morphology often adopted by cells cultivated in 2D, which may impact cell function, signaling, and interactions with other cells [88,89,90,91]. Moreover, monolayer cell culture affects the gene expression and biochemistry of the cell [83,92,93]. However, with the goal of providing more accurate data from experiments, there is a pressing need for more realistic and reliable models that can mimic the 3D aspect of the in vivo tissue [86]. 

As its name suggests, a 3D cell culture model aims to replicate as closely as possible the architectural structure of the native tissue [28]. This characteristic stands as its greatest advantage, as the results obtained are more likely to represent the expected behaviors of living organisms [83,86]. Indeed, cells developed in a 3D fashion have preserved morphology, gene expression, and cell-to-cell interaction [26,27,28,29,83,91,94,95,96]. For instance, because of the 3D nature of cells, the access to oxygen, nutrients, metabolites, and other signaling molecules varies throughout the cell due to spatial diffusion gradient, mirroring the conditions expected in vivo [84,97]. This makes it a great tool for studying various conditions such as cystic fibrosis, neurovascular and neurodegenerative diseases, cancer-initiating cells, drug transport, viral infections, and more [26,27,28,29,31,34,47,48,58,68,83,95,96]. Several 3D culture approaches have been developed such as rotating bioreactors, 3D bioprinting, organoids, organs-on-a-chip, as well as other scaffold-free and scaffold-based cultures [63,98,99,100,101]. However, these approaches are costlier and more time-consuming, as they demand specific and intricate culture conditions [28,83]. Another important consideration is the intended applications when producing a 3D model. Hence, to guarantee the engineered nasal mucosa functions as effectively as the in vivo tissue, it must closely mimic the histological traits of its native counterpart. Therefore, for drug transport and pharmacological studies, it is essential that the tight junctions of the engineered tissue closely resemble and interact with each other in a similar fashion to those of the natural tissue [32,84,87]. This similarity is vital to ensure reliable and predictable drug test results [84]. Whereas tissues intended for surgical purposes not only need to be similar histologically to their in vivo counterparts, but they must also have the ability to withstand the mechanical forces experienced by the tissue of origin [42,102]. 

In the article of Butler et al. [48], three criteria were described for the engineered nasal mucosa to be considered suitable for human transplantation. First, the tissue from which cells are to be extracted must be autologous [48]. Second, the cells must be rapidly expandable to address demanding, urgent, and critical clinical situations [48]. Third, the cells’ karyotype, expression of tissue-specific markers, differentiation, and functional capacity should closely resemble those of the tissue of origin [48]. Even if the previous criteria are met, several challenges are to be expected when developing a 3D model for engineering nasal mucosa. For example, optimizing cell extraction from biopsies is crucial to obtaining as many viable cells as possible from the native tissue. This abundance of cells is essential for subsequent use in cell culture and expansion processes [22,42,53,61]. Moreover, the nasal cavity is an area filled with pathogens. Bacterial and fungal infections are the main cause of unsuccessful cultures reported by some authors [57,69,103]. Therefore, cultivating germ-free cells isolated from nasal mucosa is a major concern in producing quality tissue-engineered nasal mucosa [69,103,104]. Indeed, the nasal cavity which serves as the tissue donor is filled with various bacteria such as *Streptococcus pneumoniae* a Gram-positive *cocci,* and *Haemophilus influenzae,* a Gram-negative coccobacillus [105]. Therefore, the usage of broad-spectrum antibiotics targeting both Gram-positive and Gram-negative pathogens is required when culturing respiratory cells. This explains why penicillin and streptomycin are the most used combinations of antibiotics in the manuscripts included in this research. However, studies did not discuss the infection rates associated with various antibiotic regimens. Moreover, several authors from the included studies combined gentamycin with their penicillin/streptomycin mixture [44,48,54,64,67,68]. Gentamycin, similarly to streptomycin is an aminoglycoside but with the ability to act against Gram-negative as well as Gram-positive bacteria, therefore, ensuring a complete coverage against bacterial infection [106]. In the cell culture literature, several authors advocate against the use of antibiotics in culturing media as they induce changes in gene expression and cell differentiation [107,108]. Moreover, a successful germ-free extraction protocol alone is insufficient for proper cell differentiation. Cells grown in the wrong media or lacking the necessary supplements will not differentiate effectively. This explains why certain authors studied the effect of different media and supplements on the growth and differentiation rate of respiratory cells [31,45,61]. These led to interesting findings, such as the supplementation of culture media with retinoic acid to ensure proper cell differentiation and ciliogenesis [42,61,109,110]. In a study led by Sachs et al., their objective was to identify the optimal media that could sustain high levels of differentiation in primary cultures of human tracheal epithelium [110]. After testing nine different media, they concluded that those with minimal or no retinoic acid supplementation resulted in poorly differentiated cells compared to those with higher concentrations (>50 nM) [110]. As a result, some of the studies included in this review supplemented their media with retinoic acid, as its beneficial effects have been established [42,61]. 

Finally, a crucial challenge to note when cultivating respiratory cells is their limited proliferative capacity in vitro [21]. Indeed, they can only be grown for 4–5 passages before reverting to a poorly differentiated phenotype [21]. Several authors have tried to work their way around this by cultivating their cells in conditions that enhance cell growth and life span [48,111]. In general, this enhanced ability to pass the cells was accomplished by reducing the degree of confluence at which the cells were passaged, transitioning from the traditional 80% to a range of 50–70%. Consequently, the increase in the number of passages was attained at the expense of amplifying the cell number at each passage. Furthermore, cell culture is highly patient-dependent, and while cells from one patient may demonstrate excellent differentiation, others may encounter challenges in the cell culture process. All these factors represent significant challenges in achieving a successful 3D bioengineered nasal mucosa.

Based on the current reviewed studies, the mechanical stress to which the nasal mucosa is subjected is particularly difficult to reproduce in vitro, especially when we think of sneezing. However, approaches using a bioreactor to impose shear stresses through airflow could be possible. As for the mechanical resistance of the tissue itself, no analysis of the mechanical properties of tissue-engineered mucosa has been described in the literature. However, many studies have been reported on the mechanical strength of tissue-engineered skin, which has been shown to be good enough to be manipulated and grafted in humans, without even requiring any special treatment [112]. Since the production of the mucosa is largely inspired by that of the skin, it is hoped that its mechanical resistance will not be an obstacle to a possible clinical translation. 

The barrier function of the nasal epithelium can be analyzed in vitro by performing absorption tests of labeled molecules, as has been done with the skin [113], in parallel to visualizing tight junction markers by immunofluorescence on tissue sections. Reproducing the immune function of the nasal mucosa is probably the greatest challenge. It is possible to incorporate monocytes/macrophages and dendritic cells into the tissue, but it is difficult to maintain their long-term survival, and especially, to ensure the maintenance of their functionality, namely, their ability to release inflammatory cytokines in response to a stimulus [114]. For this purpose, microfluidic culture systems that allow the tissue to be regularly replenished with fresh immune cells could be a smart solution, but complex to develop. The integration of lymphocytes is even more difficult, due to the need to use all cells from the same donor. However, the prospect of studying the immune response of the tissue to the deposition on the mucosal surface of different pathogens, or their antigens, would be a highly exciting evolution of this model. 

The main strength of this study is the thorough search strategy of multiple databases. This ensures a minimum risk of publication bias and enables to presentation of a complete synthesis of all the available data found in the literature. Another strength of this study is the fact that the article selection process was limited solely to articles describing a protocol for the development of the upper respiratory mucosa and not the lower as they differ histologically and functionally. As such, the findings of this review provide a thorough understanding of the upper respiratory tract cell culture techniques, but this knowledge cannot be applied to the entire respiratory tract. The main limitation of this study lies in the absence of publication of unsuccessful culture protocols, which would have been relevant for investigating the primary causes of their failure.

Current culture protocols aimed at reconstructing nasal mucosa from autologous tissue show a moderate to high degree of variability with varying degrees of success. Described optimal techniques seem to involve nasal biopsy for cell extraction, the use of antibiotics for decontamination, and ALI medium with Pneumacult for differentiation. The comparison of engineered tissues to the in vivo mucosa is predominantly based on morphological aspects. In future studies, it would be of great interest to comprehensively assess the mechanical properties of tissue-engineered nasal mucosa produced using HNEC. This holds particular significance if the bioengineered product is intended for use as a surgical replacement to the native tissue. For example, patients undergoing endoscopic endonasal skull base surgeries sometimes need a replacement flap for their native tissue. Having this tissue available would resolve a major issue for these patients.

## Figures and Tables

**Figure 1 bioengineering-11-00826-f001:**
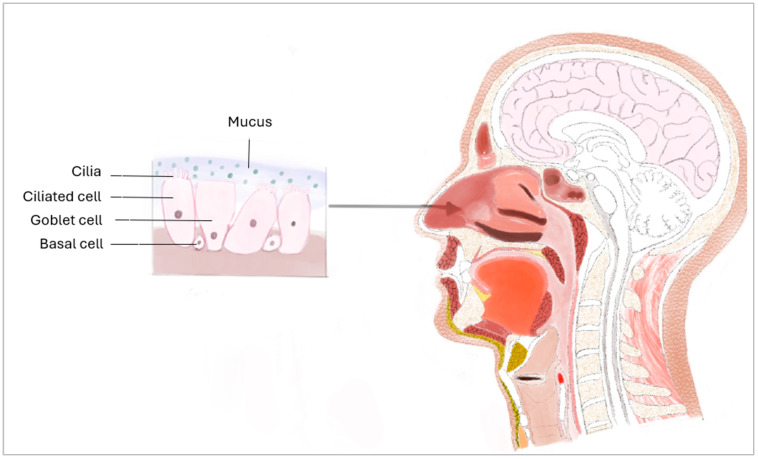
Sagittal drawing of the upper respiratory tract.

**Figure 2 bioengineering-11-00826-f002:**
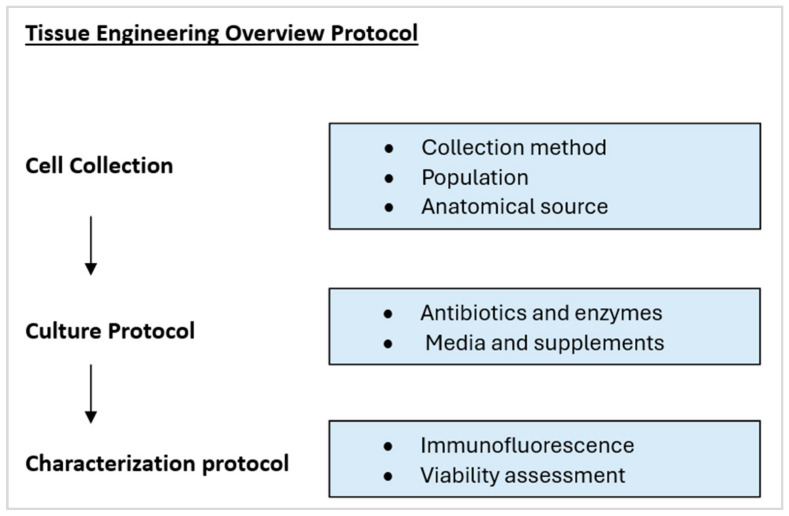
Tabular representation of tissue engineering protocol.

**Figure 3 bioengineering-11-00826-f003:**
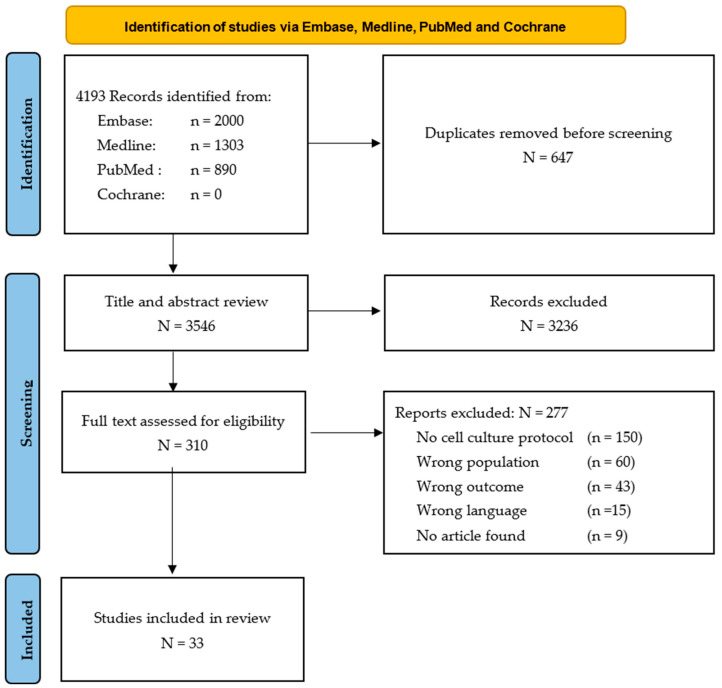
Flow diagram of article selection process from the Embase and Medline database.

**Table 1 bioengineering-11-00826-t001:** Characteristics of articles included in the review process.

REFERENCE	YEAR	CELL SOURCE	POPULATION TYPE ^a^	CELL COLLECTION	MUCOSA PURPOSE ^b^
**GOLEC ET AL. [22]**	2022	Bronchial and Nasal cells	N/A	Nasal brushing, Lung explants	Cystic fibrosis research
**MICHI ET AL. [34]**	2021	Bronchial and Nasal cells	Cadaver	Bronchial biopsy	Viral research
**LOKANATHAN ET AL. [42]**	2021	Nasal cells	N/A	Nasal biopsy	Reconstructive surgical research
**MANNA ET AL. [43]**	2021	Nasal cells	Adults	Nasal brushing	Protocol description
**LEUNG ET AL. [61]**	2020	Bronchial and Nasal cells	Cadaver	Bronchial biopsy	Physiology research
**LEE ET AL. [45]**	2020	Nasal cells	N/A	Nasal brushing	Protocol description
**KREFT ET AL. [31]**	2020	Nasal cells	N/A	N/A	Pharmacology research
**BERGOUGNAN ET AL. [44]**	2020	Nasal cells	N/A	Nasal biopsy, Bronchial brushing	Physiology research
**EVERMAN ET AL. [68]**	2018	Primary basal epithelial cells	N/A	N/A	Physiology research
**JIANG ET AL. [62]**	2018	Bronchial and Nasal cells	N/A	Bronchial biopsy	Protocol description
**GIANOTTI ET AL. [21]**	2018	Nasal cells	N/A	Nasal brushing, Lung explants	Cystic fibrosis research
**SCHAGEN ET AL. [46]**	2018	Nasal cells	Adults	Nasal brushing	Physiology research
**SCHOGLER ET AL. [47]**	2017	Nasal cells	Pediatrics	Nasal brushing	Cystic fibrosis research
**BROADBENT ET AL. [33]**	2016	Bronchial and nasal cells	Pediatrics	Bronchial brushing	Viral research
**BUTLER ET AL. [48]**	2016	Nasal cells	N/A	Bronchial biopsy	Reconstructive surgical research
**RAREDON ET AL. [63]**	2015	Bronchial and Nasal cells	Adults	N/A	Protocol description
**HUSSAIN ET AL. [49]**	2014	Nasal cells	Adults	Nasal brushing	Pharmacology research
**MULLER ET AL. [50]**	2013	Nasal cells	N/A	Nasal biopsy	Protocol description
**FULCHER ET AL. [64]**	2013	Bronchial and Nasal cells	Cadaver	Bronchial biopsy	Protocol description
**ZHAO ET AL. [51]**	2012	Nasal cells	Adults	Nasal biopsy	Physiology research
**PRYTHERCH ET AL. [65]**	2011	Bronchial and Nasal cells	Cadaver	N/A	Protocol description
**RANDELL ET AL. [54]**	2011	Nasal cells	N/A	N/A	Cystic fibrosis research
**EVEN-TZUR ET AL. [52]**	2010	Nasal cells	N/A	Nasal biopsy	Protocol description
**NORRUDDIN ET AL. [53]**	2007	Nasal cells	N/A	Nasal biopsy	Reconstructive surgical research
**CHOE ET AL. [58]**	2006	Bronchial cells	N/A	N/A	Protocol description
**WIDDICOMBE ET AL. [67]**	2005	Bronchial and Nasal cells	Cadaver	Bronchial biopsy	Protocol description
**BALS ET AL. [66]**	2004	Bronchial and Nasal cells	N/A	Bronchial biopsy	Cystic fibrosis research
**PAQUETTE ET AL. [59]**	2003	Bronchial cells	N/A	Bronchial brushing	Protocol description
**YOO ET AL. [32]**	2003	Nasal cells	N/A	Nasal biopsy	Pharmacology research
**ROSS ET AL. [55]**	1997	Nasal cells	N/A	Nasal brushing	Reconstructive surgical research
**WERNER ET AL. [57]**	1995	Nasal cells	N/A	Nasal biopsy	Pharmacology research
**STEINSVAG ET AL. [56]**	1991	Nasal cells	Pediatrics	Nasal biopsy	Protocol description
**LECHNER ET AL. [60]**	1985	Bronchial cells	Cadaver	Bronchial biopsy	Protocol description

**N/A** describes: Not available ^a^ Pediatric population describes patients <18 years old ^b^ Physiology research refers to the study of function and biological processes within the nasal mucosa. Pharmacological research refers to the study of drug transport and metabolism within the nasal mucosa.

**Table 2 bioengineering-11-00826-t002:** Cell culture protocol of included studies.

REFERENCE	2D VS 3D CULTURE ^a^	PROLIFERATIVE MEDIUM	SERUM	DIFFERENTIATION ALI MEDIUM	ANTIBIOTICS PROTOCOL	ENZYME
**GOLEC ET AL. [22]**	3D	DMEM/F12	FBS	DMEM/F12	Penicillin, streptomycin, Tazocilin, Colomycin, ciprofloxacin,	Pronase
**MICHI ET AL. [34]**	3D	DMEM/F12	FBS	Pneumacult	Penicillin, Streptomycin	Pronase
**LOKANATHAN ET AL. [42]**	3D	DMEM, DKSFM	FBS	DMEM, DKSFM	N/A	Collagenase H
**MANNA ET AL. [43]**	3D	Pneumacult	BSA	Pneumacult	Penicillin, Streptomycin	N/A
**LEUNG ET AL. [61]**	3D	BEGM	No	Pneumacult, BEGM	N/A	Protease, DNase Pronase
**LEE ET AL. [45]**	3D	DMEM	FBS	Pneumacult, BEGM, AECGM, LHC-8	Penicillin, Streptomycin	N/A
**KREFT ET AL. [31]**	3D	AECGM	N/A	AECGM	N/A	N/A
**BERGOUGNAN ET AL. [44]**	3D	DMEM/F12	FBS	DMEM, AECGM	Penicillin, Streptomycin, Gentamycin	N/A
**EVERMAN ET AL. [68]**	3D	DMEM, BEGM	BSA	Pneumacult	Penicillin, Streptomycin, Gentamycin	N/A
**JIANG ET AL. [62]**	3D	BEGM	FBS	BEGM	Penicillin, Streptomycin	Protease
**GIANOTTI ET AL. [21]**	3D	LHC-basal medium	ABS	DMEM	Penicillin, Streptomycin	Protease
**SCHAGEN ET AL. [46]**	3D	BEGM	N/A	B-ALI	N/A	N/A
**SCHOGLER ET AL. [47]**	3D	BEGM	N/A	Pneumacult	Primocin	N/A
**BROADBENT ET AL. [33]**	3D	DMEM	FBS	AECGM	Penicillin, Streptomycin	N/A
**BUTLER ET AL. [48]**	3D	DMEM, BEGM	FBS	BEGM	Penicillin, Streptomycin, Gentamycin	N/A
**RAREDON ET AL. [63]**	3D	B-ALI	N/A	B-ALI	Penicillin, Streptomycin	N/A
**HUSSAIN ET AL. [49]**	2D	AECGM	No	-	Penicillin, Streptomycin	N/A
**MULLER ET AL. [50]**	3D	BEGM	No	Pneumacult	N/A	DNase
**FULCHER ET AL. [64]**	3D	BEGM	BSA	DMEM, LHC basal medium	Penicillin, Streptomycin, Gentamycin	DNase, Protease
**ZHAO ET AL. [51]**	3D	DMEM/F12	FBS	B-ALI	Penicillin, Streptomycin	Protease
**PRYTHERCH ET AL. [65]**	3D	BEGM	N/A	DMEM, BEGM	Penicillin, Streptomycin	N/A
**RANDELL ET AL. [54]**	3D	BEGM	BSA	DMEM, LHC basal medium	Penicillin, Streptomycin, Gentamycin	Protease
**EVEN-TZUR ET AL. [52]**	3D	BEGM	No	DMEM, LHC basal medium	N/A	Pronase
**NORRUDDIN ET AL. [53]**	2D	DMEM/F12, DKSFM	FBS	-	N/A	Collagenase Protease
**CHOE ET AL. [58]**	3D	BEGM	BSA	BEGM	Penicillin, Streptomycin	N/A
**WIDDICOMBE ET AL. [67]**	2D	DMEM/LHC-9	BSA	-	Penicillin, Streptomycin, Gentamycin	Protease
**BALS ET AL. [66]**	3D	MEM	N/A	DMEM/F12	Penicillin, streptomycin, tobramycin, ceftazidime, imipenem—cilastin	Protease
**PAQUETTE ET AL. [59]**	3D	DMEM/F12	FBS	DMEM	Penicillin, Gentamycin	Collagenase H
**YOO ET AL. [32]**	2D	BEGM	BSA	-	Penicillin, Streptomycin	Pronase
**ROSS ET AL. [55]**	3D	DMEM	No	-	Penicillin, Streptomycin	Pronase
**WERNER ET AL. [57]**	2D	DMEM	FBS	-	Penicillin, Streptomycin	Protease
**STEINSVAG ET AL. [56]**	2D	DMEM	FBS	-	Penicillin, Streptomycin	N/A
**LECHNER ET AL. [60]**	2D	LHC-9	No	-	Gentamycin	N/A

**Acronyms:** ABS: Adult Bovine Serum; AECGM: Airway Epithelial Cell Growth Medium; ALI: Air Liquid Interface; B-ALI: Bronchial Air Liquid Interface Medium; BEGM: Bronchial Epithelial Growth Medium; BSA: Bovine Serum Albumin; DKSFM: Defined Keratinocytes Serum-free Medium; DMEM: Dulbecco’s Modified Eagle Medium; F12: Ham’s Nutrient Mixture; FBS: Fetal Bovin Serum; MEM: Minimum Essential Medium; N/**A**: Not available. ^a^ 2D culture refers to cells cultured in a submerged environment; a 3D culture refers to air-liquid interface culture.

**Table 3 bioengineering-11-00826-t003:** Characterization protocol of included studies.

REFERENCE	GOBLET CELLS MARKER	CILIATED CELLS MARKER	TIGHT JUNCTION MARKER	CELL VIABILITY MARKER	COMPLETE AND HANDLEABLE MUCOSA
**GOLEC ET AL. [22]**	N/A	N/A	N/A	No	No
**MICHI ET AL. [34]**	MUC5Ac	B-tubulin	ZO-1	Lactate Dehydrogenase Propidium Iodide	No
**LOKANATHAN ET AL. [42]**	N/A	N/A	N/A	No	No
**MANNA ET AL. [43]**	MUC5Ac	Acetylated tubulin	N/A	No	No
**LEUNG ET AL. [61]**	MUC5Ac	N/A	ZO-1	No	No
**LEE ET AL. [45]**	MUC5Ac	B-tubulin	N/A	Propidium Iodide	No
**KREFT ET AL. [31]**	N/A	N/A	N/A	Trypan Blue	No
**BERGOUGNAN ET AL. [44]**	MUC5Ac	Acetylated tubulin	ZO-1	Trypan Blue	No
**EVERMAN ET AL. [68]**	N/A	N/A	N/A	Trypan Blue	No
**JIANG ET AL. [62]**	N/A	N/A	N/A	No	No
**GIANOTTI ET AL. [21]**	MUC5Ac	B-tubulin	ZO-1	No	No
**SCHAGEN ET AL. [46]**	MUC5Ac	Acetylated tubulin	ZO-1	No	No
**SCHOGLER ET AL. [47]**	MUC5Ac	B-tubulin	ZO-1	No	No
**BROADBENT ET AL. [33]**	MUC5Ac	B-tubulin	ZO-1	Trypan Blue	No
**BUTLER ET AL. [48]**	MUC5Ac	Acetylated tubulin	N/A	No	Yes
**RAREDON ET AL. [63]**	MUC5Ac	Acetylated tubulin	ZO-1	No	No
**HUSSAIN ET AL. [49]**	N/A	N/A	N/A	Trypan Blue	No
**MULLER ET AL. [50]**	N/A	N/A	N/A	No	No
**FULCHER ET AL. [64]**	N/A	N/A	N/A	No	No
**ZHAO ET AL. [51]**	MUC5Ac	B-tubulin	ZO-1	No	No
**PRYTHERCH ET AL. [65]**	Periodic acid-Shift stain	N/A	ZO-1	No	No
**RANDELL ET AL. [54]**	N/A	N/A	N/A	No	No
**EVEN-TZUR ET AL. [52]**	MUC5Ac	B-tubulin	ZO-1	No	No
**NORRUDDIN ET AL. [53]**	MUC5Ac	Cytokeratin-18	N/A	No	No
**CHOE ET AL. [58]**	N/A	N/A	N/A	No	No
**WIDDICOMBE ET AL. [67]**	N/A	N/A	N/A	No	No
**BALS ET AL. [66]**	N/A	N/A	N/A	No	No
**PAQUETTE ET AL. [59]**	Periodic acid-Shift stain	N/A	N/A	No	No
**YOO ET AL. [32]**	N/A	N/A	N/A	No	No
**ROSS ET AL. [55]**	N/A	N/A	N/A	No	No
**WERNER ET AL. [57]**	N/A	N/A	N/A	Trypan Blue	No
**STEINSVAG ET AL. [56]**	Periodic acid-Shift stain	N/A	N/A	No	No
**LECHNER ET AL. [60]**	N/A	N/A	N/A	No	No

Acronyms: FITC: Fluorescein Isothiocyanate; N/A: Not available; ZO-1: Zonula-adherence-1.

## Data Availability

The data that supports the findings of this study are available in the Appendix A of this article.

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
