# Peer review of "Bioengineering Human Upper Respiratory Mucosa: A Systematic Review of the State of the Art of Cell Culture Techniques"

_bioengineering, 2024, doi:10.3390/bioengineering11080826_

Round 1

Reviewer 1 Report

Comments and Suggestions for Authors

To authors

An interesting and comprehensive perspective on the existent literature reporting on methods to engineer upper respiratory mucosa aimed at final human use.

Several comments below

Abstract- well written and informative. Please note that regenerative medicine and tissue engineering is NOT dedicated only to plastic surgery and ophthalmology how the first paragraph in the abstract (and probably the AI assistant) might seem to suggest. The application in upper respiratory tract is not difficult BECAUSE of multiple protocols, is difficult rather because of intricacy of respective tissue with immune system and perhaps is indeed associated with many methods that have been tested. Is always good not to take association as causality unless proven as such. In this reviewer perspective it would be good to underline starting from the abstract that currently no clinical or even translational application of such culture methods exist for reasons already mentioned by authors (such as mimicking morphology only but not mechanical properties).

Introduction correctly and comprehensively expose the anatomy (not too much the function, this could be improved with one two phrases regarding mucosa role in breathing and immune defense) Several major causes of disfunction and/or damage are listed.

Good enough definition of tissue engineering however it lacks mentioning the main purpose that is to produce implantable/transplantable organ/tissue bioequivalents for treating a lost, damaged or absent organ. The role of TE for producing such bioequivalent organ/tissue to serve as models of disease with or without patient specific cells should be mentioned .

Based on literature screening following PRISMA reccoemndation, the authros further describe criteria for literature search, study selection and inclusion within the review. In the followings the respective methods for culturing components of uppoer respiratory tract are described in detail

This reviewer would point pout the necessity of more clearly stressing out methods for read outs of the respective culture (a graphic or a tabular form) as they are of most interest for the readers.

What is further lacking is the description of so called “protocols for further use in pneumology, virology” announced in the abstract. Have this methods been used by respective authors to generate in vitro tissue models if yes, which are the studies and how were the models used?

Has any of this studies succeeded in obtaining an implantable/transplantable tissue in animal models of a respiratory disease of some kind? If yes please describe, if not, please comment possible causes for which such development has not been performed

A short subchapter describing challenges in reproducing in vitro complex mechanical stress, barrier function and particularly the interface and intricate functioning with local and systemic immune components should be introduced. Where stands the situation of reproducing functionality in this respect, from what current reviewed studies is concerned? Can this approach be improved and if yes what would be the methods (co culture, interface generating dishes, bioreactors, microfluidics, organ on a chip?)

Reviewer 2 Report

Comments and Suggestions for Authors

The review article presents a comprehensive review of the current cell culture protocols of human upper respiratory cells and mucosal tissues. The authors also pointed out the need for further research on the mechanical properties of engineered tissues. There are a few minor issues that the author can address:

1. The authors could add some discussion on unsuccessful culture protocols, as these can provide valuable insights into the field.

2. The authors mention the need for future research on the mechanical properties of tissue-engineered nasal mucosa, it would be helpful to include a brief overview of any existing studies.

3. The author can include more specific examples of how tissue-engineered nasal mucosa is being used in practice.
